

# The behavior patterns of giraffes (*Giraffa camelopardalis*) housed across 18 US zoos

Jason D. Wark and  Katherine A. Cronin

Animal Welfare Science Program, Lincoln Park Zoo, Chicago, IL, United States of America

## ABSTRACT

Interpreting animal behavior in the context of welfare can be inherently challenging given the limited behavior data available for many species housed in zoos. Describing common behavior patterns may help animal managers by providing additional background when assessing the individuals in their care. Although valuable, these efforts require a large, collaborative approach and have, consequently, been rare. Here, we share the behavior patterns of zoo-housed giraffes, an iconic and commonly housed megafauna in zoos. Behavior data were evaluated for 66 giraffes living across 18 AZA-accredited zoos using the ZooMonitor Community platform. Data were recorded during 10-minute observation sessions. Observations were conducted during daytime hours over the course of approximately one year at each zoo (mean total observed time per individual = 23.2 hr). The most common behaviors observed were feeding/foraging behaviors, which accounted for 38.6% of the mean visible time budget across giraffes. Time spent in these behaviors varied by individual and ranged from 14.3% to 69.3% of visible time. Stereotypic behaviors occurred in all study individuals, with oral stereotypic behaviors being most common. Although prevalent, stereotypic behaviors varied considerably across giraffes, with some individuals exhibiting these behaviors only on a few occasions to an individual that exhibited these behaviors once every few minutes. This study provides a robust evaluation of giraffe behavior across zoos to present a picture of their common behavior patterns in managed care. We hope these multi-institutional behavior patterns can provide perspective to aid animal managers in evaluating giraffes in their care.

## INTRODUCTION

Understanding common patterns of animal behavior in managed care is likely of interest to many animal care managers, as a departure from typical patterns may have welfare implications. However, defining typical patterns of behavior can be challenging. Collecting relevant data to make inferences on the regularity of a behavior pattern can be challenging. Behavior, by definition, is a dynamic process that enables animals to respond to changes in their environment and is reinforced through individual experiences (*Gomez-Marin & Ghazanfar, 2019*; *Levitis, Lidicker & Freund, 2009*). Thus, behavior patterns may vary between individuals and can change over time. Systematic monitoring of animal behavior

Corresponding author
Jason D. Wark, jwark@lpzoo.org

**How to cite this article** Wark JD, Cronin KA. 2024. The behavior patterns of giraffes (*Giraffa camelopardalis*) housed across 18 US zoos.
*PeerJ* **12**:e18164 http://doi.org/10.7717/peerj.18164

at a single institution can help shed light on an individual animal's behavioral repertoire and patterns. However, classifying these individual behavior patterns as common or typical requires the additional context of the behavioral patterns of other individuals of the same species housed at additional institutions.

One approach to interpreting behavior patterns is to compare the behavior of an individual in managed care to their wild conspecifics (*e.g.*, *Veasey, Waran & Young, 1996b*; *Melfi & Feistner, 2002*; *Miller, Chas & Hacker, 2016*; *LaDue et al., 2022*). This approach largely draws from a philosophical view that the performance of natural behaviors is fundamental for welfare and the presence of behaviors not occurring in the wild are "abnormal" and detrimental to welfare (*Hill & Broom, 2009*; *Browning, 2020*). This view has remained a central tenet in animal welfare science since the introduction of the Five Freedoms and the fifth freedom of "Freedom to express normal behavior." As *Hill & Broom (2009)* state, normal behavior in this view describes those behaviors occurring in healthy animals living in environments that provide unrestricted behavioral opportunities appropriate for the species. Unfortunately, animals in the wild do not live in a utopia and face corresponding challenges, albeit different ones than typically faced in captivity. Therefore, animals in the wild are unlikely to express an idealistic behavioral pattern. Although comparisons to wild animals can provide some guidance, aiming to replicate the behavioral expression observed in the wild in captive settings, without more detailed attention to the function and context of behavior, has been challenged by several authors (*Veasey, Waran & Young, 1996a*; *Špinka, 2006*; *Browning, 2020*; *Cronin & Ross, 2020*; *Bartlett, Grinsted & Freeman, 2023*).

An additional, and possibly more relevant, comparison is that between the behavior of a zoo-housed animal with many other zoo-housed animals of the same species. This approach still has limitations; the range of behavior observed will depend upon the individuals and the conditions in which they live. However, this perspective can shed light on the potential for behavioral expression in the managed population. This comparison necessitates a multi-institutional approach. Although others have raised the importance of multi-institutional research (*Swaisgood & Shepherdson, 2005*; *Watters, Margulis & Atsalis, 2009*; *Whitham & Wielebnowski, 2013*), the logistical challenges of recruiting for and coordinating these studies to ensure systematic and reliable data collection have made them relatively rare, but some notable examples exist. In their expansive study on elephant welfare, *Meehan et al. (2016)* explored the impacts of the physical environment, social experience, and husbandry practice on Asian and African elephants that included behavior data from 89 animals housed at 39 zoos. One finding from their research was that stereotypic behaviors, which may be considered abnormal from a natural behavior view, were quite common in zoo-housed elephants, occurring in 85% of the animals studied (*Greco et al., 2016*). Stereotypic behaviors were also found to vary considerably across individuals, with some animals exhibiting these behaviors as little as 0.5% of their time while others exhibited these behaviors for up to 68% of their time. Findings from this study have been incorporated into the Association of Zoos and Aquariums (AZA) Standards for Elephant Management and Care, a set of guidelines required for AZA Accreditation. More recently, multi-institutional research in zoos and aquariums has revealed species-typical behavior

patterns for bottlenose dolphins (*Lauderdale et al., 2023*) and chimpanzees (*Whitham et al., 2023*) as well.

Giraffes are one of the most commonly housed species of megafauna in zoos, with more than 500 individuals living across more than 100 zoos accredited by the Association of Zoos and Aquariums. Currently accepted taxonomy recognizes one species of giraffe with nine subspecies, although this has been disputed and some have argued for four species of giraffe to be recognized (*Fennessy et al., 2016*). Giraffe subspecies managed in the AZA include the Masai, reticulated, Rothschild, and resulting hybrids of these subspecies. Giraffes are browsers and, as the tallest animal, are adapted to feed on leaves and branches in the tree canopy, spending a majority of their day foraging and feeding in the wild (*Pellew, 1984*; *Du Toit & Yetman, 2005*; *Gitau et al., 2024*; but see *Paulse et al., 2023*; *Deacon, Smit & Grobbelaar, 2024*).

There has been a long history of multi-institutional research on giraffes. In their pioneering work, *Veasey, Waran & Young (1996b)* were the first to conduct a multi-institutional study on zoo-housed giraffes, comparing the behavior of individuals housed across four different zoos to each other and giraffes in the wild. Since then, more multi-institutional studies of giraffes have been conducted, including a review of oral stereotypic behavior (*Koene & Visser, 1997*); a comparison of female giraffe behavior (*Bashaw, 2011*), evaluations of guest feeding programs (*Orban, Siegford & Snider, 2016*; *Ramis et al., 2022*), and an assessment of seasonal habitat changes (*Razal, Bryant & Miller, 2024*). Despite this work, our understanding of species-typical behavior patterns remains incomplete as these past studies have been conducted at a small number of organizations (*Veasey, Waran & Young, 1996b*; *Koene & Visser, 1997*; *Bashaw, 2011*; *Razal, Bryant & Miller, 2024*), on a specific demographic group (*Bashaw, 2011*), or during a limited time of the year (*Orban, Siegford & Snider, 2016*). However, past work has highlighted potential behavioral concerns for giraffes, including limited opportunities for browsing and the presence of stereotypic behaviors (*Veasey, Waran & Young, 1996b*; *Koene & Visser, 1997*; *Baxter & Plowman, 2001*; *Bashaw et al., 2001*; *Tarou, Bashaw & Maple, 2003*; *Bergeron et al., 2006*; *Fernandez et al., 2008*; *Bashaw, 2011*; *Orban, Siegford & Snider, 2016*; *Okabe et al., 2022*; *Depauw, Verbist L & Salas, 2023*; *Walldén, 2023*; *Razal, Bryant & Miller, 2024*).

Here, we contribute new information to the question of giraffe behavioral repertoires in North American zoos. Given previous research, we pay particular attention to feeding and stereotypic behaviors in giraffes housed across 18 AZA-accredited zoos. This exploratory study provides a broad overview of species-typical behavior patterns in giraffes in managed care and provides important benchmarks for future inquiries regarding how individual behavior relates to population-level patterns.

## MATERIALS & METHODS

### Subjects and housing

Focal subjects included 67 giraffes (26 males, 41 females) housed across 18 US zoos that were accredited by the Association of Zoos and Aquariums (AZA). One female giraffe passed away shortly after the start of data collection and was excluded from analysis, resulting

in 66 giraffes being considered in this study. This study was reviewed and approved by institutional research review boards at each zoo. These zoos represented organizations of varying size and were geographically located across the United States. Each participating zoo was asked to observe, where possible, a minimum of three giraffes from their herd. The selection of focal subjects was pseudorandomized such that the primary investigator (PI) provided an initial random selection of three focal individuals to each zoo who was then given the option to include additional individuals based on their monitoring capacity or swap individuals based on their management priorities. The number of focal animals at each zoo ranged from two to eight individuals.

Prior to the start of the study, surveys were administered to participating zoos to gather information on individual and habitat characteristics. Although these data were not analyzed in the current study, we present this summary information here to provide a thorough overview of the study animals and their husbandry. Focal giraffes in this study ranged in age from 1 to 29 years, with a median age of 9 years. The total herd size (including non-focal individuals) ranged from 2 to 16 individuals, with a median herd size of five giraffes. Giraffes were primarily managed in a single social group (n = 59) and housed socially overnight (n = 61). During periods of the year with outdoor access, roughly half of the study giraffes were shifted into indoor areas overnight (n =32). Most giraffes in the study were not contracepted (n =43). Of the focal subjects, 49 giraffes were reported prior to the start of the study to exhibit a stereotypic behavior, with oral stereotypies being the most commonly reported (n = 39), followed by locomotor (n = 17), head rolling (n = 9), and self-injurious (n = 1).

Habitat size varied widely across the zoos, with the smallest habitat measuring 464 m$^2$ and the largest habitat measuring 263,045 m$^2$. The median habitat size was 3,507 m$^2$. The percent of the total habitat space that was outdoors ranged from 4% to 100%, with a median of 90%. The percent of the total habitat space that featured soft substrate ranged from 75% to 100%, with a median of 92%. Most zoos housed giraffes with other species (n = 14). This most commonly included other artiodactyl species (n = 13) or birds (n = 9).

## Data collection

Data collection and project coordination were conducted through the ZooMonitor Community collaborative platform. A project was created in ZooMonitor (Version 5; Lincoln Park Zoo, 2024) and shared to the Community platform prior to the start of data collection. Participating zoos were then able to view and join the project in the Community and access the project's ethogram, behavior sampling methodology, and training materials. After joining the project, participating zoos were then instructed to add their focal individuals, animal habitat maps, and observers to their project. Each zoo also completed short surveys on their individuals and habitats in ZooMonitor. Behavior data were recorded at each zoo using the ZooMonitor app (*Wark et al., 2019*) and shared with the PI through the Community feature of ZooMonitor.

Data were collected for approximately one year at each participating zoo, starting in January, 2022 and continuing to March, 2023. As several focal giraffes were added during the study and one individual passed away near the end of data collection, a full year of

data collection (*i.e.,* minimum of 45 weeks of data collection) was not possible for some individuals (n = 6). These individuals were included in the analysis to provide the most comprehensive view of giraffe behavior patterns possible. Behavior observations were conducted during daytime hours (6:00 to 18:00) and were approximately balanced across morning (12:30 or before: 4,973 sessions, 54%) and afternoon (after 12:30: 4,231 sessions, 46%) time periods. The number of sessions recorded each hour varied and was lowest before 9:00 and after 15:00 and highest during the early morning (10:00 and 11:00; see Fig. S1 for a histogram of sessions by hour).

Giraffe behavior was recorded during 10 min. observation sessions at each zoo. Observers were primarily volunteers and interns but did include researchers and animal keepers. Observers were instructed to record data for a single focal animal during observation sessions but, in a small number of cases (3.5% of sessions), multiple focal animals were observed simultaneously. The ethogram of behaviors observed in this study is shown in Table 1. All behaviors on the ethogram were recorded using instantaneous point-sampling at one-min. intervals. In addition, all occurrences of stereotypic behaviors were noted.

To ensure reliable and consistent observations within and across zoos, a three-part observer testing process was conducted. This process relied heavily on video materials and was informed from past research that identified potential gaps to live, in-person reliability testing (*Wark, Wierzal & Cronin, 2021*). First, to familiarize observers with the appearance of ethogram behaviors, observers were administered a 20-question online test that featured brief video snippets of different behaviors to identify. After completing this test, observers then began inter-observer reliability testing. They were then asked to complete two 10-min. video reliability tests, with a mean percent agreement of 85% or better required to pass. Observers that did not pass initially were given two additional attempts (*i.e.,* six total video reliability tests maximum). If an observer did not pass video reliability testing (n = 4), they were not permitted to record data for this project. Observers that passed video reliability tests were then required to complete two in-person reliability tests at the giraffe habitat with a project lead at each zoo. All project leads had prior experience with research and/or giraffes and had also completed the first two virtual testing parts. Given logistical challenges, it was not possible for project leads to conduct in-person reliability tests across institutions. All occurrences data from one observer were excluded from analysis due to errors in the recording protocol.

## Data analysis

To provide an account of the range of behavioral expression of zoo-housed giraffes, the percent of time (*i.e.,* percent of intervals) an individual was engaged in each behavior was first calculated for each observation session. Then, an overall mean percent of time was calculated for each focal giraffe and behavior across sessions. As the number of recorded intervals varied, sessions with less than five intervals recorded were excluded from the analysis to prevent artificially inflated percentages (*Wark et al., 2023*). Data are presented for both the individual behaviors and combined behavioral categories. To illustrate the variability within behavioral categories, the standard deviation of the mean (SD), the coefficient of variation (CV) (SD/ mean * 100), the range (max −min), and interquartile

**Table 1   Ethogram.**

| Behavior name | Behavior category | Definition |
|---|---|---|
| Standing | Inactive | Animal is upright with weight supported on feet and not performing another behavior listed. |
| Sitting[a] | Inactive | Animal has weight supported on legs or ventral surface. May be alert with head elevated or sleeping with head resting on their body. |
| Browsing/ Feed | Feed/ Forage/ Drink | Animal is using tongue or mouth to strip or pluck leaves or bark from a branch (can include environmental foliage as well as diet items). This includes chewing and consumption of food items gained through browsing. |
| Extractive Foraging/ Feed | Feed/ Forage/ Drink | Animal is using tongue or mouth to extract food from within an enclosed object (e.g., hanging extractive feeding bags or buckets). This includes chewing and consumption of food items gained through extractive foraging. |
| Ruminating | Feed/ Forage/ Drink | Regurgitation and chewing cud of previously eaten food. Does not include periods of chewing which might accompany foraging and should be recorded as "Feeding." |
| Other Feeding/ Drinking | Feed/ Forage/ Drink | Animal is performing any other feeding behavior (e.g., feeding from troughs, grazing on grass, foraging across substrate, guest hand feeding). |
| Locomotion | Locomotion | Animal is moving at least one body's length in a non-stereotypical manner. |
| Tongue Play[b] | Stereotypy | Animal is moving tongue outside of mouth in a repetitive, twisting or rolling movement. May have food item present but not actively chewing food. |
| Repetitive Licking[b] | Stereotypy | Animal is repeatedly moving tongue across a non-food, stationary object (e.g., walls, fencing, or trees). |
| Pacing[b] | Stereotypy | Animal is walking in a repetitive manner along a fixed path without an apparent goal or function. The animal must move along the path three times to qualify as pacing. [Note: If an interval occurs during the first two transects and the animal continues into a pacing bout, score pacing]. |
| Other Stereotypy[b] | Stereotypy | Animal is performing any other non-functional, invariant, and repetitive behavior not listed above (please score whether the stereotypy type is Oral, Motor, Locomotor, or Other). |
| Other Solitary | Other Solitary | Animal is performing any other solitary behavior, including but not limited to self-maintenance behaviors, exploratory behaviors, and elimination behaviors. |
| Affiliative | Social | Animal makes physical contact with another conspecific individual in an affiliative manner, including rubbing necks, heads, bodies, or muzzles or sniffing and licking the muzzle or non-anogenital area of the body. |
| Sexual | Social | Animal is physically mounting or attempting to mount a conspecific animal or investigating the animal or environment in a sexual manner (e.g., anogenital exam, urine investigation, flehmen). |
| Agonistic | Social | Animal performs any aggressive behavior, either with or without contact, or any displacement/ avoidance behavior. |
| Other Social Behavior | Social | Animal is performing a social behavior not previously listed. |
| Behavior Obscured | Not Visible | The behavior of the animal cannot be determined but the location of the animal is known and in the habitat spaces under observation (i.e., record a corresponding space use location). |
| Animal Not Visible | Not Visible | The animal is completely not visible and its location is unknown (i.e., do not record a space use location) or in an off-exhibit area not under observation. |

**Notes.**
[a] This behavior is comparable to "Rest (Lying)" of *Seeber, Ciofolo & Ganswindt (2012)*.
[b] These behaviors were recorded on an all-occurrence and interval basis.
range (IQR; 75th–25th quartile) were calculated. In addition to absolute measures of variability (*i.e.,* SD, range, IQR), the CV was included as a relative measure of variability to aid comparison between common and rare behaviors, as this metric standardizes the variability relative to the mean.

For stereotypic behaviors that were recorded on an all-occurrence basis, a rate was calculated by dividing the sum of the number of occurrences during each observation session by the number of visible intervals during a session. These visible behavior rates per session were then averaged to calculate an overall mean rate of time for each individual. As with the analysis of interval data, sessions with less than five visible intervals were excluded from analysis. Rates were calculated for both individual behaviors and combined behavioral categories based on the type of stereotypy (*i.e.,* locomotor, oral, motor).

Analyses and visualizations were performed using R statistical software (version 4.3.1; *R Core Team, 2023*).

## RESULTS

A total of 9,204 focal observation sessions were analyzed. The total time an individual was observed ranged from 3.3 to 62.0 hr, with a mean total time observed of 23.2 hr. The mean number of weeks that focal individuals were observed was 50.1 and ranged from 13 to 58.1 weeks. The mean number of observation sessions per focal individual was 139.5 sessions and ranged from 20 to 373 sessions.

### Species-typical behavior patterns

The range of behavioral expression observed in this study of 66 giraffes is shown in Figs. 1 and 2, with summary statistics for each behavior category displayed in Table 2. Giraffes in this study spent the largest portion of their time visible engaged in a feeding or foraging behavior ($=X = 38.6\%$): browsing ($=X = 13.2\%$, SD $= 9.7\%$); other feeding/foraging ($=X = 13.1\%$, SD $= 8.4\%$); extractive foraging ($=X = 12.3\%$, SD $= 8.5\%$). Standing was the most common behavior observed ($=X = 17.3\%$, SD $= 7.0\%$) followed by ruminating ($=X = 15.6\%$, SD $= 6.7\%$). For most individuals, stereotypic behaviors constituted a relatively appreciable portion of the overall visible time budget ($=X = 10.4\%$, SD $= 5.5\%$): repetitive licking ($=X = 4.8\%$, SD $= 8.6\%$); tongue play ($=X = 4.8\%$, SD $= 9.2\%$); pacing ($=X = 0.5\%$, SD $= 1.5\%$); other stereotypy ($=X = 0.4\%$, SD $= 0.9\%$). Most behaviors occurred at a consistent level between 9:00 to 15:00 (Fig. 2). However, greater variation in behavior was noted before and after these times when fewer observations were conducted (Fig. S1).

As a category, feeding and foraging behaviors showed the largest absolute variation across individuals, with an interquartile range (IQR) of 17.6% of visible time and range of 55.0% of visible time from the maximum to minimum observed per individual (Table 1). However, when evaluating the relative variability, adjusting for differences in the mean percent of time between behavior categories, feeding/ foraging behaviors had the lowest CV, as these behaviors were the most common. In contrast, stereotypic behaviors and social behaviors displayed large variation across individuals, with CVs over 100% (*i.e.,* standard deviation was greater than the mean, Table 2). Inactive, locomotion, and ruminating

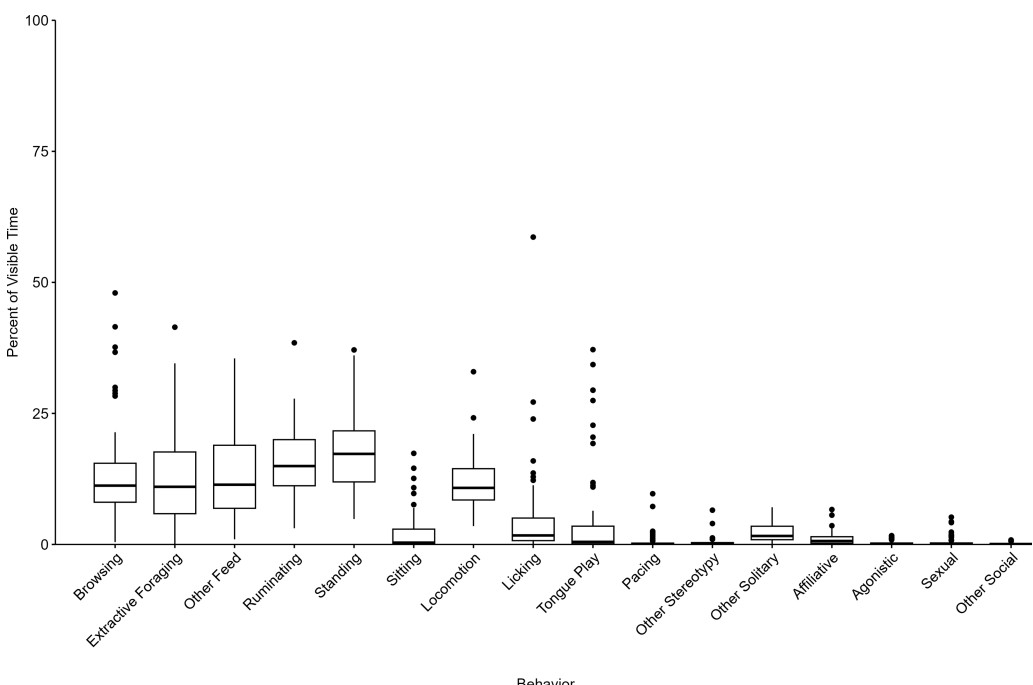

**Figure 1** **The behavior patterns of giraffes observed in this study.** The boxplot displays the percent of visible time for each individuals mean behavior value as boxes representing the 25th and 75th percentiles, the median indicated as a horizontal line, whiskers representing the largest value within 1.5 times the interquartile range, and dots to indicate individual outliers defined as values above and below 1.5 times the interquartile range.

behaviors showed a similar level of variation across study individuals, with CVs of 41.3%, 42.2%, and 43.1%, respectively.

Giraffes in this study were rarely out of view of the observers (not visible: $\bar{X} = 5.3\%$, SD $= 5.9\%$).

## Feeding/ foraging behavior

The individual variation in visible time spent engaged in feeding or foraging behaviors is shown in Fig. 3. The maximum visible time spent feeding by a giraffe in this study was 69.3%. This individual was also observed to spend the most time browsing of any giraffe ($\bar{X} = 48.0\%$, SD $= 36.3\%$). The minimum visible time spent feeding or foraging by a giraffe was 14.3%. All giraffes were observed browsing or extractive foraging and most giraffes engaged in both behaviors (63/66 individuals).

## Stereotypic behavior

A total of 5,976 occurrences of stereotypic behavior were observed in this study. All giraffes in this study were observed to exhibit a stereotypic behavior at least once. Individual stereotypic expression ranged from individuals that exhibited only one occurrence of a stereotypic behavior during the study (three individuals) to an individual that exhibited 715 occurrences of stereotypic behavior.

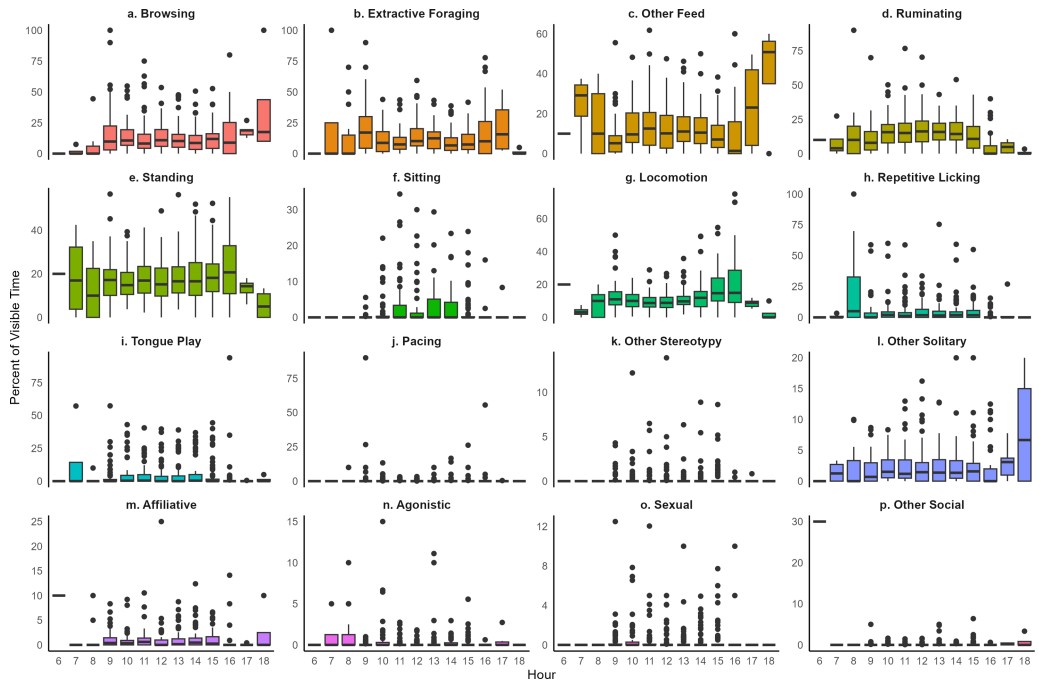

**Figure 2 The hourly behavior patterns of giraffes observed in this study.** The boxplots display the percent of visible time for each individuals mean behavior value as boxes representing the 25th and 75th percentiles, the median indicated as a horizontal line, whiskers representing the largest value within 1.5 times the interquartile range, and dots to indicate individual outliers defined as values above and below 1.5 times the interquartile range. Note the difference in y-axes scales for each behavior.

**Table 2 Summary statistics of the percent of visible time giraffes spent engaged in different behavior categories.**

| Behavior category | Mean (SD) | Median | Range | IQR[a] | CV[b] |
|---|---|---|---|---|---|
| Feeding/ Foraging/ Drinking | 38.6 (13.1) | 37.9 | 14.3–69.3 (55.2) | 29.1–46.6 (17.6) | 34.2% |
| Ruminating | 15.6 (6.7) | 15.0 | 3.1–38.5 (35.4) | 10.6–19.4 (8.8) | 43.1% |
| Inactive | 19.5 (8.0) | 19.2 | 5.1–42.1 (37.0) | 14.4–23.9 (9.5) | 41.3% |
| Locomotion | 11.7 (5.0) | 10.8 | 3.5–33.0 (29.5) | 7.8–13.8 (6.0) | 42.2% |
| Stereotypy | 10.4 (11.7) | 5.5 | 0–59.6 (59.6) | 0.05–11.0 (11.1) | 111.5% |
| Other Solitary | 2.2 (1.8) | 1.6 | 0–7.1 (7.1) | 0.3–2.9 (2.6) | 79.8% |
| Social | 1.9 (2.2) | 1.0 | 0–11.0 (11.0) | 0.05–2.0 (1.9) | 112.0% |

**Notes.**
[a]IQR, Interquartile range.
[b]CV, Coefficient of Variation.

The most common type of stereotypic behavior observed was oral, which accounted for 89.0% of the stereotypic behavior occurrences. Motor stereotypic behaviors were the next most frequent (7.1% occurrences) and locomotor stereotypic behaviors were the least frequent (3.9%). The majority of individuals exhibited more than one type of stereotypic behavior (40/66 individuals). However, of those individuals, most had

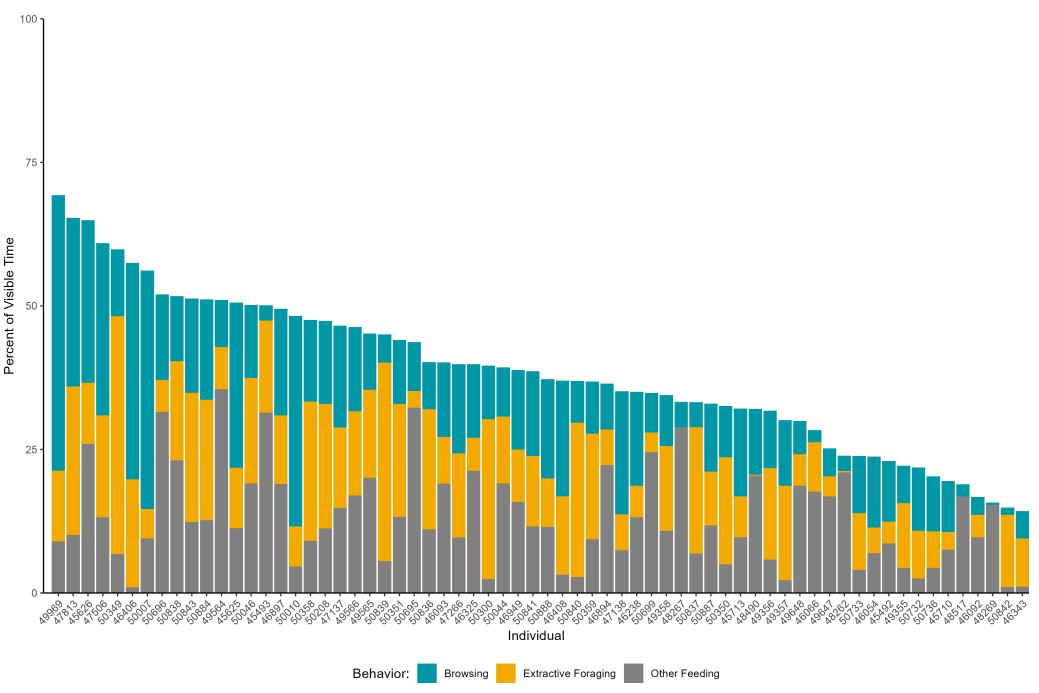

**Figure 3** **The percent of visible time spent in feeding or foraging behaviors across individual giraffes.** The x-axis shows anonymized giraffe IDs.

a dominant stereotypy that accounted for more than 90% of their occurrences (24/40 individuals) (Fig. 4). Some hourly variation in stereotypies was noted (Fig. 5).

Although stereotypic behaviors were observed throughout the population, the time invested in stereotypic behavior varied greatly. Rates of stereotypic behaviors ranged from 0.00080 to 0.33 occurrences per minute, which corresponds to once every 1,250 min to once every 3 min of observation time.

## DISCUSSION

The goal of this study was to broadly describe common giraffe behavior patterns in US zoos to better understand what typical behavior may look like in managed care and identify abnormal patterns that may signify a welfare concern. Through collaborative data collection across 18 zoos, this study provides a broad view of the behavior patterns of zoo-housed giraffes. In addition to a general overview of behavioral expression, we examined foraging and stereotypic behaviors in detail, given the past attention towards these behaviors and their potential relationship to welfare (*Bergeron et al., 2006*).

For most maintenance behaviors, such as feeding and foraging, ruminating, inactivity, and locomoting, variation across individuals was low when considering the relative variability of common and rare behaviors (*i.e.,* CV), presenting a clear picture of typical activity in AZA-accredited zoos. Social behaviors showed a high degree of variation when considering both absolute and relative measures of variability, however, overall rates were quite low despite differences in herd sizes across the study population. Stereotypic behaviors,
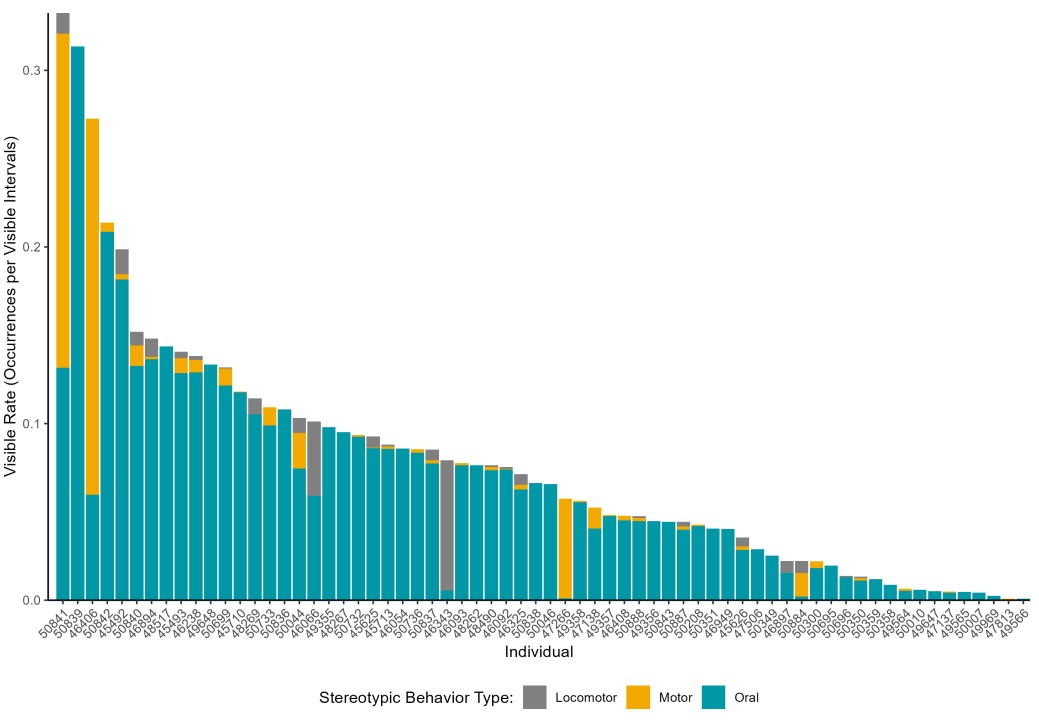

**Figure 4** **The mean visible rate of stereotypic behavior types across individual giraffes.** The x-axis shows anonymized giraffe IDs.

on the other hand, showed a high degree of variation between individuals and were more common and observed to a varying degree in all individuals. The expression of stereotypic behaviors, therefore, may be considered typical for giraffes living in AZA-accredited zoos but their frequency may depend on individual- and organization-level factors.

Similar to past research, we found feeding behaviors represented a large portion of the visible time budget of giraffes (mean 39% of time and IQR of 29.1% –46.6%). Notably, this estimate is generally higher than has been reported in previous multi-institutional studies (based on published values or extracted from graphs). For example, *Veasey, Waran & Young (1996b)* found giraffes at four UK zoos spent between 17% to 26% of their time feeding and foraging, with the mean time across zoos of approximately 23%. *Koene (2013)* similarly reported giraffes at four Dutch zoos spent between 12% to 27% with a mean time across zoos of approximately 19%. *Bashaw (2011)* observed feeding behaviors by female giraffes ranged from approximately 17% to 41% across three herds, with a mean time feeding of approximately 27%. Similarly, *Orban, Siegford & Snider (2016)* reported giraffes across nine zoos feeding for approximately 20% of time. *Gussek et al. (2018)*, in their study of giraffe nutrition across 12 German zoos, observed giraffe feeding for 30% of time. Although the time spent feeding in the present study is higher than past multi-institutional studies, some single-institution studies have reported even higher levels of feeding (*e.g.*, *Schüßler, Gürtler & Greven, 2015*: 48% of time; (*Depauw, Verbist L & Salas, 2023*: 43.4% of time; *Walldén, 2023*: 64% of time), highlighting the range of time spent feeding that may be

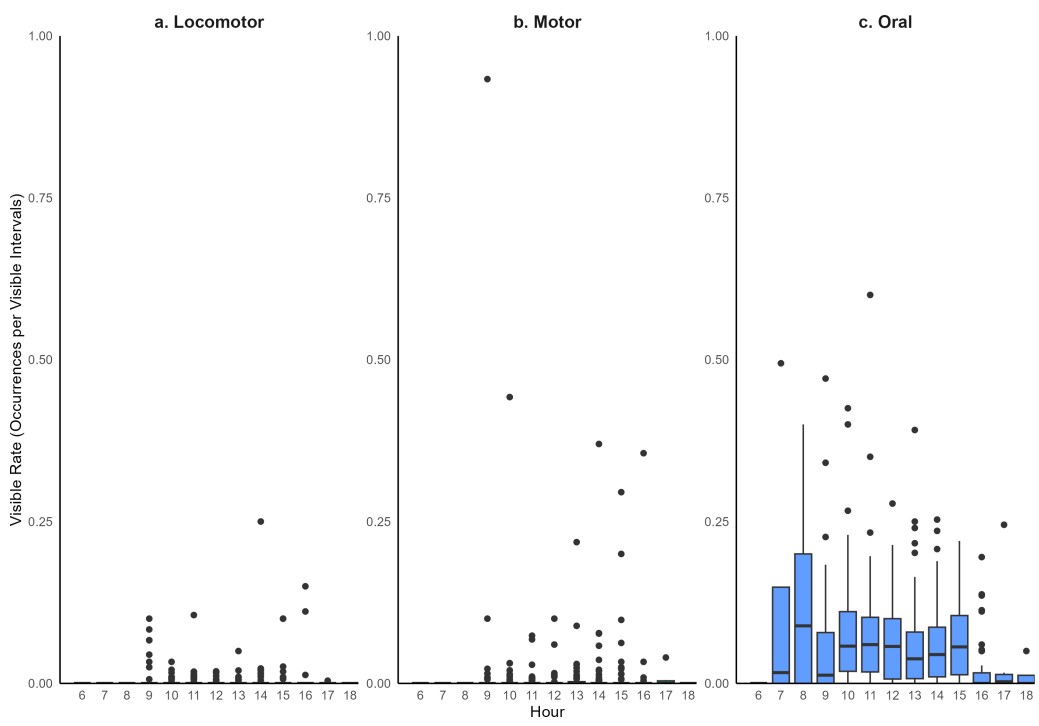

**Figure 5** **The hourly visible rate of stereotypic behavior types of giraffes observed in this study.**

possible in managed care. Understanding the range of time giraffes in zoos spend feeding and foraging can be a valuable tool for animal managers evaluating individuals in their care and the potential for behavioral changes. It is generally agreed that more time spent feeding and foraging can support giraffe welfare (*Rose, 2023*); combining that background knowledge with the current data would suggest attention is warranted for individuals on the low end of these zoo-based estimates (*i.e.,* feeding and foraging less than 30% of their visible time budget), and that it is within the potential of zoos to support more feeding and foraging time by resident animals.

The greater time feeding observed in the present study compared to prior research may reflect greater attention to feeding behaviors in giraffes as a result of past research and husbandry recommendations (*e.g.,* *Bergeron et al., 2006*; *EAZA Giraffe EEPs, 2006*). In the present study, giraffes spent approximately equal amounts of time engaged in each of the three feeding behaviors recorded: browsing, extractive foraging/ feeding, and other feeding. Many zoos have been striving to increase browsing and extractive foraging opportunities to prolong feeding bouts (*Fernandez et al., 2008*) and it is encouraging to see these behaviors well represented in the behavioral profiles of giraffes in this study. In a recent example, *Depauw, Verbist L & Salas (2023)* evaluated changes in how giraffes were fed at a zoo that included an emphasis on increased browse and use of slow feeders, among other dietary changes, and found giraffes nearly doubled the amount of time spent feeding and foraging (24.5% of time before *vs* 43.4% after) and used their tongues more during feeding bouts. *Walldén (2023)* observed a similar increase in time spent feeding after diets were fed

exclusively through slow feeders. *Kearney, Ball & Hall (2024)* noted increased feeding in giraffes fed a processed high-fiber diet. In addition to the previous studies, others have examined the nutritional needs of giraffes and argued for an increase in fiber and browse in diets (*Baxter & Plowman, 2001*; *Clauss et al., 2002*; *Hatt et al., 2005*; *Gussek et al., 2018*). Some zoos in the current study may have already begun this journey and implemented similar changes, yielding higher estimates for time spent feeding and foraging.

Unfortunately, stereotypic behaviors, which are generally indicative of current or past welfare compromise, were observed in all the study animals. This prevalence is comparable to what has been reported in past behavior studies of giraffes (*e.g.*, *Orban, Siegford & Snider, 2016*: 93% of study animals) but higher than what might have been expected based on past survey research. In a previous survey that included 214 giraffes and 29 okapis, *Bashaw et al. (2001)* found stereotypic behaviors occurred in 80% of giraffes and okapis. In the present study, a survey conducted before data collection commenced found project participants reported stereotypic behaviors in only 71% of the study animals. Although these reports broadly correspond and highlight the prevalence of stereotypic behaviors in giraffes, it is important to note that specific estimates of behavior prevalence may vary based on the study methods. Similar discrepancies between surveys and data collection have been observed in reports on the prevalence of stereotypic behaviors in chimpanzees (c.f., *Birkett & Newton-Fisher, 2011*; *Jacobson, Ross & Bloomsmith, 2016*), emphasizing the value of systematic data collection over retrospective reports when possible. Surveys may often be chosen for their simplicity, however, even short-term data collection may be sufficient and superior to surveys in some cases, as the prevalence in stereotypic behaviors observed by *Orban, Siegford & Snider (2016)* from three days of intensive data collection was comparable to data recorded sporadically over the course of a year from the present study.

The stereotypic behaviors observed in the current study were primarily oral stereotypies, corresponding to what others have previously reported in giraffes (*Bashaw et al., 2001*; *Bergeron et al., 2006*; *Orban, Siegford & Snider, 2016*). In the present study, giraffes spent approximately 10% of their daytime budget performing stereotypic behaviors, a similar amount to what has been reported previously (*Veasey, Waran & Young, 1996b*: 10–21%; *Baxter & Plowman, 2001*: 2–18%; *Bashaw, 2011*: 9–14%; *Orban, Siegford & Snider, 2016*: 13.9–18.3%; *Razal, Bryant & Miller, 2017*: 3.5–8%;) but higher than has been reported by some (*Gussek et al., 2018*: 4.7% for oral stereotypies). However, the rate of stereotypic behaviors varied greatly across giraffes, with some individuals rarely exhibiting stereotypic behaviors, most exhibiting them at a moderate level similar to what has been reported previously (*Orban, Siegford & Snider, 2016*), and several individuals exhibiting these behaviors regularly.

Taken together, these results suggest some signs of progress in addressing concerns surrounding giraffe behavior. Encouragingly, the overall time spent feeding appeared higher than most past reports in zoos. However, given the widespread prevalence of stereotypic behaviors, additional work is needed. As past studies have found the rate of oral stereotypies may be related to overall time spent feeding and foraging (*Koene & Visser, 1997*; *Orban, Siegford & Snider, 2016*; *Duggan, Burn & Clauss, 2016*), continued and

increased efforts to promote browsing and extractive foraging may be warranted (*e.g.,* *Depauw, Verbist L & Salas, 2023*; *Fernandez et al., 2008*; *Walldén, 2023*). Ultimately, this may suggest that giraffes in zoos would experience better welfare, at least as measured by stereotypic behavior, if they spend a similar amount of time feeding and foraging as their wild conspecifics, which has been estimated at 50–75% of the time budget (*Pellew, 1984*; *Du Toit & Yetman, 2005*; but see *Paulse et al., 2023*; *Deacon, Smit & Grobbelaar, 2024*). Some have suggested that increasing the time spent ruminating may be more important than feeding when trying to address oral stereotypies (*Baxter & Plowman, 2001*), although more work is needed to confirm this relationship. Unfortunately, as others have noted, stereotypic behaviors, once established, can be difficult to eliminate (*Garner, 2006*). Thus, it will be important to determine a realistic goal for individuals currently expressing stereotypies and it may be prudent to focus on avoiding the emergence of stereotypic behavior in recently born individuals. For example, although *Depauw, Verbist L & Salas (2023)* observed a large increase in time spent feeding by giraffes after a series of husbandry changes, only one individual was observed to significantly decrease their time performing repetitive licking behaviors. In another study, increasing the hay-to-grain ratio in the diet of giraffes was found to decrease tongue play oral stereotypies but did not change repetitive licking oral stereotypies (*Monson et al., 2018*). In their study on the use of slow feeders, *Walldén (2023)* documented a consistent decrease in stereotypic behaviors across the majority of the giraffes in their study, suggesting a strategy for reducing these behaviors but likely not eliminating them altogether. More work is needed to understand the perseverative nature of stereotypic behavior in giraffes and their responsiveness to husbandry and dietary changes.

While this study makes a valuable contribution to our understanding of giraffe behavior, and behavioral potential, across US zoos, it raises several important questions about how behaviors are impacted by husbandry and environment, as well as how behaviors are impacted by one another. For example, did habitat size impact behavior, as the largest giraffe habitat in this study was 566 times bigger than the smallest? Was there an effect of sex on time spent feeding, as others have found (*Young & Isbell, 1991*)? This work is currently underway and will hopefully shed light on specific predictive factors influencing behaviors of interest that can aid managers in making evidence-based decisions to enhance welfare. Finally, this study did not consider dietary influences on behavior or the potential anticipatory nature of stereotypic behaviors. We encourage future studies to consider these factors (c.f., *Gussek et al., 2018*).

This was the first study of the ZooMonitor Community platform. This collaborative feature in ZooMonitor introduces new tools for facilitating multi-institutional research, making it possible for researchers to publish their projects to a shared space visible to ZooMonitor users around the world. Researchers can then manage their studies through built-in tools in the ZooMonitor Community. The need for multi-institutional research has been highlighted by others (*Swaisgood & Shepherdson, 2005*; *Watters, Margulis & Atsalis, 2009*; *Whitham & Wielebnowski, 2013*) and, with the widespread use of the ZooMonitor app in zoos and aquariums around the world, the ZooMonitor Community has the potential to

increase collaborative research and accelerate our collective knowledge of normal behavior patterns for the many species housed across zoos and aquariums.

## CONCLUSIONS

Understanding typical behavior patterns can aid zoos and aquariums in identifying normal and, consequently, abnormal behavior of animals in their care. Here, we evaluated the behavior of 66 giraffes across 18 zoos, providing a robust account of their behavior patterns under current husbandry conditions. Consistent with past research, feeding and foraging behaviors were the most frequently observed behaviors. Given the focus of zoos on promoting these natural behaviors, it was encouraging to see they occurred even more frequently than has been generally reported in past zoo research. Unfortunately, also consistent with past research was the prevalence of stereotypic behaviors, particularly oral stereotypies. Large inter-individual variation in stereotypic behavior was noted, suggesting there may be specific individual- or institutional-level factors in the housing or care of giraffes contributing to these behaviors. Additional research is underway to explore these factors in more detail. This study was conducted using new collaborative research features in the ZooMonitor behavior recording app. More multi-institutional research is needed to build our collective knowledge of normal behavior patterns for species housed in zoos and aquariums. We encourage others to consider these new tools and advance these efforts for more species.

## ACKNOWLEDGEMENTS

This project would not have been possible without the 17 zoos and many amazing colleagues that supported Lincoln Park Zoo in coordinating data collection for this project, including Cincinnati Zoo and Botanical Garden (Catherine Razal, David Orban), Cleveland Metroparks Zoo (Noah Dunham), Columbus Zoo and Aquarium (Adam Felts, Shannon Borders), Dallas Zoo (Alex Hauenstein, Allison Rackley), Denver Zoo (Ali Young, Heather Genter, Jessica Meehan), Detroit Zoo (Grace Fuller, Megan Jones), Disney's Animal Kingdom (Andrew Alba), Jacksonville Zoo and Gardens (Aimee McDonnell, Lindsay Mahovetz-Myers), North Carolina Zoo (Emily Lynch), Oklahoma City Zoo (Kimberly Leser), Oregon Zoo (Laurel Westcott), San Diego Zoo (Jessica Sheftel), San Diego Zoo Safari Park (Louisa Radosevich), San Francisco Zoo (Bethany Krebs), Toledo Zoo (Beth Posta), Woodland Park Zoo (Bonnie Baird), and Zoo Atlanta (Marieke Gartner). In addition, we thank the more than 80 observers that generously helped record data for this project. We would also like to thank Sherri Horiszny and Amy Schilz for their support of this project on behalf of the AZA Giraffe Species Survival Plan. We are grateful to Maureen Leahy, Dave Bernier, Mike Murray, and Cassandra Kutilek for their support in facilitating this project at Lincoln Park Zoo. Lastly, we extend immense thanks to Natasha Wierzal for her help in developing training materials and helping coordinate data collection at Lincoln Park Zoo.

## Funding

This work was supported by the Institute of Museum and Library Services (MG-245613-OMS-20). The funders had no role in study design, data collection and analysis, decision to publish, or preparation of the manuscript.

## Grant Disclosures

The following grant information was disclosed by the authors:
The Institute of Museum and Library Services: MG-245613-OMS-20.

## Competing Interests

The authors declare there are no competing interests.

## Author Contributions

- Jason D. Wark conceived and designed the experiments, performed the experiments, analyzed the data, prepared figures and/or tables, authored or reviewed drafts of the article, and approved the final draft.
- Katherine A. Cronin conceived and designed the experiments, authored or reviewed drafts of the article, and approved the final draft.

## Data Availability

The raw data are available in the Supplementary Files. Data were pseudonymized to protect the identity of participating institutions.

## Supplemental Information

Supplemental information for this article can be found online at http://dx.doi.org/10.7717/peerj.18164#supplemental-information.

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
