# Peer review of "The behavior patterns of giraffes (Giraffa camelopardalis) housed across 18 US zoos"

_PeerJ, doi:10.7717/peerj.18164_

## Round 0.1 · original submission · Major Revisions

Dear authors, please carefully correct all the reviewers' comments. Try to formulate your thoughts as strictly as possible. Laconicism should not hinder a creative approach to analyzing your data. The article is very interesting and I hope that it will become one of the most cited articles in this area of ​​research.

Reviewer 1 ·

Basic reporting

This manuscript is clear, well-written, and well-sourced. Tables and figures are clear and raw data are provided. The research is descriptive, so no hypotheses are needed. This does appear to be an appropriate unit of publication (with the additions suggested in #3); the authors indicate they are preparing a separate manuscript on individual and institutional predictors of feeding and stereotypies, which seems like an appropriate way to partition their large dataset.

Experimental design

Data collection protocol is clear and technically sound. Training procedures for observers - as well as how clearly they are reported - are exemplary. This is very rare in a multi-institutional study and the authors should be commended! It would be valuable to add notes in the ethogram (Table 1) indicating where behavioral definitions depart from the published giraffe ethogram: Seeber, P. A., Ciofolo, I., & Ganswindt, A. (2012). Behavioural inventory of the giraffe (Giraffa camelopardalis). BMC research notes, 5, 1-9.

Validity of the findings

The findings are robust and presented using appropriate descriptive statistics. Figures and tables are clear and well-designed. Conclusions are clearly stated, but could be strengthened by examining and controlling for diurnal and seasonal patterns.
Giraffe behavior varies significantly by time of day and season both in situ (e.g., Deacon et al, 2024; Paulse et al, 2023) and in human care (e.g., Tarou et al., 2000; Razal et al., 2017). Times of day and dates for each observation are included in the supplemental data; observations are not evenly distributed across times of day (e.g., 10a-12p are the most common observation times - these coincide with morning feedings in many institutions). The authors should use these data to add a description of "normal" diurnal and seasonal behavior patterns and/or statistically control for time of day and observation month/season when they report "normal" activity budgets. Also, it would be helpful to know if giraffes' oral stereotypies occurred most commonly at a particular time of day, as is common for other feeding-related stereotypies like regurgitation and reingestion (e.g., Cassella, C. M., Mills, A., & Lukas, K. E. (2012). Prevalence of Regurgitation and Reingestion in Orangutans Housed in North American Zoos and an Examination of Factors Influencing its Occurrence in a Single Group of Bornean Orangutans: Prevalence of Orangutan R/R in North American Zoos. Zoo Biology, 31(5), 609–620. https://doi.org/10.1002/zoo.21000), and whether controlling for observation times/months would reduce the observed individual differences in these behaviors.

Additional comments

In interpreting their findings, the authors should include discussion of the possibility that the diet options provided to captive giraffe in accredited zoos are not adequate to support their health beyond Monson et al.'s study. They should consider whether feeding giraffes primarily high-fibre leafy greens might improve health, increase feeding time, and reduce or eliminate oral stereotypies in giraffe as it does in another browser, the gorilla (Less, E. H., Bergl, R., Ball, R., Dennis, P. M., Kuhar, C. W., Lavin, S. R., Raghanti, M. A., Wensvoort, J., Willis, M. A., & Lukas, K. E. (2014). Implementing a low-starch biscuit-free diet in zoo gorillas: The impact on behavior. Zoo Biology, 33(1), 63–73. https://doi.org/10.1002/zoo.21116.)
Some sources on giraffe nutrition and health that might be valuable for such a discussion are:
Clauss, M., Lechner-Doll, M., Flach, E. J., Wisser, J., & Hatt, J. M. (2002). Digestive tract pathology of captive giraffe. A unifying hypothesis. Proceedings of the European Association of Zoo and Wildlife Veterinarians, 4, 99-107. https://www.zora.uzh.ch/id/eprint/3542/10/EAZWV_giraffe_hypothesis_2002V.pdf
Gussek, I., Große‐Brinkhaus, C., Südekum, K. H., & Hummel, J. (2018). Influence of ration composition on nutritive and digestive variables in captive giraffes (Giraffa camelopardalis) indicating the appropriateness of feeding practice. Journal of animal physiology and animal nutrition, 102(2), e513-e524.
Gussek, I., Hirsch, S., Hartmann, M., Südekum, K. H., & Hummel, J. (2017). Feeding practices for captive giraffes (Giraffa camelopardalis) in Europe: a survey in EEP zoos. Journal of Zoo and Aquarium Research, 5(1), 62-70.
Hatt, J. M., Schaub, D., Wanner, M., Wettstein, H. R., Flach, E. J., Tack, C., ... & Clauss, M. (2005). Energy and fibre intake in a group of captive giraffe (Giraffa camelopardalis) offered increasing amounts of browse. Journal of Veterinary Medicine Series A, 52(10), 485-490.
Kearney, C. C., Ball, R. L., & Hall, M. B. (2024). Effects of altering diet carbohydrate profile and physical form on zoo‐housed giraffe Giraffa camelopardalis reticulata. Journal of Animal Physiology and Animal Nutrition. Advance online publication. https://doi.org/10.1111/jpn.13957

·

Basic reporting

This study reports on behavioural observations in North American zoo giraffes. By providing the activity budgets of many individual zoo giraffes, it provides a data basis for future exploration of correlates of these activity budgets.

In my opinion, this is important work. I found it highly interesting when reading.

But at first I felt a bit irritated by self-promoting sentences, which, due to the repetitiveness and obstinacy of the rhetorics changes into my main impression when reading the manuscript: that of a self-promoting approach – also due to omissions of facts about the method employed. Whereas the former is at the discretion of the authors and editors (following a common fashion nowadays of using self-promoting language in scientific texts (Vinkers et al. 2015) that means that scientific readers need to constantly blend out parts of the material they read), the latter must be addressed in more detail.

I have made comments on the attached pdf. I list the main points here in the following, but ask the authors to go through the pdf nevertheless.

The raw data is not shared - I think this is a precondition for PeerJ.

Language:
Self-promoting language
The authors state, quite repetitively:
l. 29 ‘This study is the largest evaluation of giraffe behavior across zoos and provides the most complete picture of their species-typical behavior patterns in managed care’
l. 111 ‘To our knowledge, this is the largest study of giraffe behavior in zoos’
l. 249 ‘this study provides the strongest insight yet into behavior patterns of zoo-housed giraffes’
l. 365 ‘providing the most complete picture of their species-typical behavior patterns to date.’

In my view, such language is not scientific but self-promotion, and scientific readers need to blend out these sentences (that have no factual value, do not add to the data reported, nor to the interpretation).
For me, these sentences create the impression that (i) the authors think they need to say this (rather than letting the reader decide on non-scientific questions like this one), (ii) that the authors think this is an important asset (bespeaking their own concept of science), (iii) that the authors do not trust the readers’ attention spans.
I am aware that such language is fashionable nowadays.

I want to address some of these statements:

The authors omit (I do not believe deliberately, just by accident) some giraffe literature, including (Baxter and Hansson 2001, Hummel et al. 2006, Schüßler et al. 2015, Duggan et al. 2016, Gussek et al. 2018, Gitau et al. 2024, Kearney et al. 2024). Among these, Gussek et al. claim to have observed in 12 zoos (less than the 18 of the present study) 95 animals (more than the 60+ of the present study). Now, which one is the ‘larger’ study? Why care? Why make such a statement? But making such a ‘larger to our knowledge’ statement without thoroughly checking the giraffe literature leaves a strange taste. My standard recommendation is: don’t make superlative statements about your work, and then you won’t make such mistakes. And Gussek et al. are specific about the time in relation to feeding events. Again, this is not a competition, unless you decide to make it one. But then you have to go all the way.

Without any explanation about the spread across the day and the relation to feeding times, the present study cannot claim to give the most ‘complete’ picture.

The present study presents the activity budgets without any explanatory or exploratory connections to diet, enclosure size, animal factors etc. (these are announced as ‘in progress’ in l. 340). Without such correlations / explanations, the present work hardly ‘provides the most complete picture of their species-typical behavior patterns in managed care’ or ‘the strongest insight yet’. Actually, any study that EXPLAINS giraffe behaviour delivers stronger insight. Again, for me, this is a strange approach to science. As a scientist, I do not try to provide STRONGER insights than someone else. As a scientist, I strive to provide strong insights. Full stop. If that insight is stronger than that of someone else should be mainly up to the reader. I would always hope my insights are complementary to that of others, or maybe corrective, or new, but STONGER? Such language is fashionable, but as a reviewer, if the recommendation to avoid such a competitive concept of science is not followed, then the reasons why these insights are stronger than those provided by others must be listed explicitly. Why is a study that does not provide any correlates of stereotypical behaviour of ‘stronger insight’ than the many studies that assess correlates of stereotypical behaviour, for example? I would say any approach, even with lower n, that provides cues as to what might be a correlate of stereotypies provides stronger insights for zoo managers to adjust their husbandry. The present study does not deliver any such insight. Again: this is not a problem, the lack of such ‘insights’ – the paper is cool and great without it. But if the authors claim to give the insights when they are not there, then the writing and presentation style becomes questionable.

Other language issues:
In the Intro, there is a passage about the definition of ‘normal’. The authors themselves use the word ‘common’ with respect to elephant work l. 77. This is the appropriate word for describing behaviours that occur a lot. They are ‘common’. One could also call them ‘frequent’. But ‘normal’ has a strong normative connotation that should not be used if just based on frequency without additonal ideas what defines the norm. This is less of a problem as I first thought when reading the Intro, as the defined use of the word ‘normal’ is quite rare anyhow in the mansucript. Nevertheless, I recommend to adopt the word ‘common’ or ‘frequent’ for a description behaviors that occur frequently.

l. 61 there is a general statement ‘Although these comparisons to wild animals may provide some insight into the potential range of behavioral expression for a species, their use as a benchmark for welfare in captive settings has been challenged’ that is correct, I recommend a more detailed view. Because some benchmarks are actually considered relevant by people (such as feeding or rumination times for ruminants), whereas evidently some others not (predator avoidance, intraspecific aggression). Summatively questioning all ‘benchmarks’ from natural habitats is, in my view, too tendentious.

l. 211 10% of visible time spent in stereotypies is not ‘modest’ in my view. Throughout the manuscript, it appears to me that the language used with respect to stereotypies is quite defensive. One does not have to rhetorically bash this, but one also does not need to be so defensive. Together with the omission of the details of observation time and observation periods, this creates a picture of a study designed to absolve zoos. I think it would serve our zoo scene better if the tone would be more objective.

l. 296 please add ‘only’ as indicated in pdf

m clauss

References cited:
Baxter R, Hansson L (2001) Bark consumption by small rodents in the northern and southern hemispheres. Mammal Rev 31:47-59.
Duggan G, Burn C, Clauss M (2016) Nocturnal behaviour in captive giraffe (Giraffa camelopardalis) – a pilot study. Zoo Biology 35:14-18.
Gitau CG, Mbau JS, Ngugi RK, Ngumbi E, Muneza AB (2024) Activity budget and foraging patterns of Nubian giraffe (Giraffa camelopardalis camelopardalis) in Lake Nakuru National Park, Kenya. Ecology and Evolution 14:e11463.
Gussek I, Große‐Brinkhaus C, Südekum K-H, Hummel J (2018) Influence of ration composition on nutritive and digestive variables in captive giraffes (Giraffa camelopardalis) indicating the appropriateness of feeding practice. Journal of Animal Physiology and Animal Nutrition 102:e513-e524.
Hummel J, Clauss M, Baxter E, Flach EJ, Johansen K (2006) The influence of roughage intake on the occurrence of oral disturbances in captive giraffids. Pages 235-252 in Fidgett, A, Clauss, M, Eulenberger, K, Hatt, J, M, Hume, I, Janssens, G, and Nijboer, J, editors. Zoo animal nutrition III. Filander Verlag, Fürth, Germany.
Kearney CC, Ball RL, Hall MB (2024) Effects of altering diet carbohydrate profile and physical form on zoo‐housed giraffe Giraffa camelopardalis reticulata. Journal of Animal Physiology and Animal Nutrition 108:1119-1133.
Schüßler D, Gürtler WD, Greven H (2015) Aktivitätsbudgets von Rothschildgiraffen (Giraffa camelopardalis rothschildi) in der „Zoom Erlebniswelt Gelsenkirchen“. Der Zoologische Garten NF 84:61-74.
Vinkers CH, Tijdink JK, Otte WM (2015) Use of positive and negative words in scientific PubMed abstracts between 1974 and 2014: retrospective analysis. BMJ 351:h6467.

Experimental design

This work was done using many different observers. Observer compatibility was ensured very diligently in an approach described in l. 166-179. This part is very good, in my view.
Most other aspects of the method remain unclear.
Who were the observers? This is a crucial information (see below). Volunteers, curators, keepers? This should be explained at least by descriptive statistics. The data should be evaluated against this, e.g. whether animals observed by keepers had a systematically different activity budget than animals observed by volunteers.

These observers observed the animals for 10 min time slots over the course of one year. How should such an approach be represented, e.g. in the abstract? The authors choose to represent this as (l. 24) a one-year observation period. However, in l. 202-204, the number of 10 min sessions per animal indicate that on average, an animal was observed for 23 hours, with a range of 4-60 hours per animal. The fact that the authors do not say this themselves (but just mention the number of observation sessions, which sounds more impressive), does not feel right.

The average and range of actual time animals were observed must be represented, including in the abstract.
23 hours corresponds to 3 days of daytime observations. This is, in its scope, significantly different from the meaning of ‘one year observation period’.
Spreading a total of 23 h of observations over a time period of one year does not make the results more comprehensive, or more reliable, but more questionable. I would trust an activity budget made during 3 consecutive 8 h observations (3 days) much more compared to many 10 min intervals spread across a whole year. This should be discussed.
With many individual 10 min sessions, the question how these were spread across the day, and across the other activities performed by the observers, becomes paramount and must be represented in much more detail than the one sentence in l. 158 ‘were approximately balanced by time of day’.

Please split the day in one-hour slots (from 7:00 to 18:00) and display, for the average and range, the proportion of observed time represented by these slots per animal. This should then enable you to have a measure of ‘evenness of spread’ (e.g., the CV when doing the average of all hour-slots) of observations across the day, which should be tested against the results. I recommend to do the same for ‘morning-noon-afternoon-evening’ (7-9, 10-12, 13-15, 16-18). This kind of information should be readily available to the authors.
By the way, the company AKONGO offers to create activity budgets in exactly the same way as the present study, and they spend particular attention to covering the whole day equally across all 10 min sessions – this is a major task (and reason why to leave this organisation to a company), compared to zoo personnel choosing their 10 min sessions at convenience.
A clear hypothesis is that with a more biased spread of observation sessions, there is more average feeding activity.

Who did the observations? If it were keepers on duty (which I would find the most logical solution), it becomes important when they did the observations in relation to their work schedule. From my experience with zoo keeper teams, I would expect them to do the first round of observation once they finished their morning tasks, which includes feeding. Or after they finished their afternoon tasks, which includes feeding … the point is: in order to trust the results, one would want to know whether the majority of 10 min sessions occurred directly after feeding times. How many occurred before a feeding event (i.e., with the leftovers from the previous feeding)? The hypothesis that the excellent results are due to this effect should be addressed in the data evaluation (if that kind of information is available, which it might well not be) or must at least be stated in the discussion.

m clauss

Validity of the findings

see above - the validity of the findings cannot be assessed without the additional checks / information mentioned above under 'methods'.

m clauss

Additional comments

l. 137 please state that an analysis of these factors in relation to the behavior data will be done at a later stage

Fig. 2 recommend other colors (think of red-green difficulty)
Fig. 3 recommend to use same y-axis unit as Fig. 2 to make it comparable to other studies; as it is, in cannot be compared

Table 1: ‘sitting’ is typically used for a posture where the front legs are in an upright position with the ground; I would refer to this here as ‘lying’
please add ‘feeding from hay rack’ to the ethogram in the category where you had it (or were there no hay racks in these zoos?)
as a comment – pacing should ideally always be separated into anticipatory and non-anticipatory (no problem if not possible in hindsight)

m clauss

---

## Round 0.2 · accepted · Accept

The article is interesting, I hope it will interest a wide range of specialists and cause a lively discussion.